# Growth and Decay of Fecal Indicator Bacteria and Changes in the Coliform Composition on the Top Surface Sand of Coastal Beaches during the Rainy Season

**DOI:** 10.3390/microorganisms11041074

**Published:** 2023-04-20

**Authors:** Soichiro Tamai, Hiroshi Shimamoto, Kei Nukazawa, Yoshihiro Suzuki

**Affiliations:** Department of Civil and Environmental Engineering, Faculty of Engineering, University of Miyazaki, Miyazaki 889-2192, Japanshimamoto@cc.miyazaki-u.ac.jp (H.S.);

**Keywords:** beach, fecal indicator bacteria, microbiota, coliform, *Enterobacter*

## Abstract

High counts of bacteria are present in beach sand, and human health threats attributable to contact with sand have been reported. In this study, we investigated fecal indicator bacteria in the top surface sand of coastal beaches. Monitoring investigations were performed during a monsoon when rainfall occurs randomly, and the composition of the coliforms was analyzed. The coliform count in the top surface sand (depth < 1 cm) increased by approximately 100 fold (26–2.23 × 10^3^ CFU/100 g) with increasing water content because of precipitation. The composition of the coliforms in the top surface sand changed within 24 h of rainfall, with *Enterobacter* comprising more than 40% of the coliforms. Estimation of factors that changed the bacterial counts and composition revealed that coliform counts tended to increase with increasing water content in the top surface sand. However, the abundance of *Enterobacter* was independent of the sand surface temperature and water content. Coliform counts in the top surface sand rapidly increased and the composition showed remarkable variations because of the supply of water to the beach following rainfall. Among them, some bacteria with suspected pathogenicity were present. Controlling bacteria in coastal beaches is important for improving public health for beachgoers.

## 1. Introduction

Coastal beaches are crucial for recreational areas such as beaches, yacht harbors and activities including surfing. However, waterborne diseases caused by pathogenic microorganisms have become a problem on beaches, aside from rivers and lakes [1,2,3,4]. Therefore, the safety of beachgoers is a concern in the US and Europe [5,6,7,8]. In the US, monitoring investigations and the development of detection methods for fecal indicator bacteria, which are indicators of the presence or absence of pathogenic microorganisms, are being actively promoted [9,10]. In 2018 data, fecal indicator bacterium counts in 2580 sites in 4253 swimming areas in the US exceeded the standard on at least 1 day during the bathing season. In addition, at 546 of these sites, the excessive fecal indicator bacterium counts were observed on more than 25% of the sampling days [11]. The EU uses *Escherichia coli* (*E. coli*) and *Enterococcus* counts to assess bathing water quality [12]. In 2021, 88.0% of the EU coastal bathing sites had excellent water quality [13]. Conversely, official water quality investigations of beaches in Japan are performed only twice annually before the bathing season. Few investigations have been conducted during or after the bathing season. However, fecal coliforms and *streptococci* have been detected at high concentrations in coastal areas during the rainy season and summer. In particular, bacterial counts were reported to rapidly increase after rainfall [14,15,16,17,18]. Furthermore, sandy beaches are likely to have 10–100-fold higher bacterial counts than seawater [19], and a positive relationship exists between the time spent on the beach and the risk of disease [20]. Therefore, the impact of contact with beach sand on human health is an issue [21,22]. However, few continuous investigations of fecal indicator bacteria in the top surface sand of beaches, which are most likely to come in contact with humans, have been conducted. The top surface of beach sand becomes dry, the temperature rises markedly, and the water content declines on sunny days. Contrarily, the water content of beaches increases rapidly upon rainfall. The relationship between soil water content and bacterial growth has already been reported, and water content is the most important factor controlling bacterial growth [23]. However, few studies have examined sand on coastal beaches.

Therefore, this study was conducted at two recreational beaches, namely, Kizaki Beach and Shirahama Beach, in Miyazaki, Japan to elucidate the factors controlling the increase and decay of fecal indicator bacterium counts in the top surface sand and clarify the causal relationship with precipitation and water content in the sand. Miyazaki is located on the Pacific side of Kyushu, and it has a temperate climate. The investigation focused on the rainy season, which is hot and humid, and continuous monitoring was conducted for 2 months. We focused on coliforms, the counts of which fluctuate markedly before and after rainfall, and examined the composition of the coliforms during the rainy season. We also conducted a simulated experiment in a laboratory using sand from beaches and investigated changes in the coliform counts associated with the addition of water to the sand. In addition, factors and trends for changes in bacterial counts were estimated via statistical analysis.

## 2. Materials and Methods

### 2.1. Water Sample Collection

Figure 1 shows a map of the area surrounding each beach. The investigation was conducted at Kizaki Beach, an international surfing beach facing the Pacific Ocean in Miyazaki, and Shirahama Beach, which is adjacent to Kizaki Beach. The Kiyotake River (drainage area: 166.4 km^2^; length: 28.8 km) is located north of Kizaki Beach, and the Kaeda River (drainage area: 53.8 km^2^, length: 17.5 km) is located north of Shirahama Beach. The investigation was conducted from 23 May 2016 to 26 July 2016, including the rainy season (rainy season in Miyazaki: 4 June–18 July; maximum daily precipitation during the rainy season: 252.2 mm on 8 July). Appendix A shows data (sunshine duration, wind speed, precipitation, and air temperature) from meteorological stations near the sampling site. From 2016 (the year of the survey) to March 2023 (the present time), the study area has not been altered by human activities such as civil engineering and river improvement projects. The coastal environment is stationary. Therefore, the information obtained from this survey can be utilized even today.

At each beach, a shovel was used to collect samples less than 1 cm below the surface (surface area, approximately 1–2 m^2^) at the upper high tide shoreline (supratidal zone) where beachgoers rest, as these areas are normally unaffected by waves and tides. This sample was designated the top surface sand (supratidal top surface sand). In this study, we focused on the top surface sand, and investigations were conducted almost daily for 40 sampling sessions. The top surface sand between the edge of the surf and the high tide shoreline (intertidal zone) and the sand 10 cm below the surface in the supratidal zone and intertidal zone were collected every 1–2 weeks; these samples were labeled as intertidal top surface sand, supratidal 10 cm sand, and intertidal 10 cm sand, respectively. The intertidal 10 cm sand at the surface was collected from three random locations using acrylic columns in the area where the top surface sand was collected, and the composite samples were analyzed. In addition, seawater samples were collected at depths of approximately 30–50 cm from the edge of the surf of each beach. The temperature of each sample was measured in the field with a bar thermometer. Sampling required 20–30 min. Samples were returned to the laboratory for immediate analysis. The travel time from our university to Kizaki Beach and Shirahama Beach was approximately 10–20 min.

### 2.2. Bacteria Counting Methods

The membrane filter method was used to count coliforms and *E. coli* [24]. In total, 5 g of sand were placed in a centrifuge tube. Then, 40 mL of sterile saline solution was added, and the mixture was stirred vigorously by hand for 2 min. Then, 40 mL of the supernatant solution was filtered through a membrane filter (0.45-µm pore size, Advantec, Tokyo, Japan). The filter was applied to CHROMagar ECC medium (Chromagar, Paris, France), a selective medium for coliforms and *E. coli*, and incubated at 37 °C ± 0.5 °C for 24 h. For samples expected to have high bacterial concentrations because of rainfall, the supernatant solution was diluted 10–100 fold, and 10 mL of the diluted solution were filtered. For seawater, 100 or 1 mL of seawater was passed through a membrane filter and applied to CHROMagar ECC medium as described for sand samples. After incubation, mauve colonies on the filter were designated as coliforms, and blue colonies were designated as *E. coli*. Mauve colonies formed on the filter were isolated on brain heart infusion (BHI) medium (Becton, Dickinson and Company, Bergen, NJ, USA) and incubated at 37 °C ± 0.5 °C for 24 h. A maximum of 30 coliforms-positive colonies were isolated from each sample. If 30 isolates could not be isolated, all coliforms-positive colonies were isolated.

*Enterococci* were counted via the membrane filter method. Each sand sample (5 g) was placed in a centrifuge tube, 40 mL of phosphate-buffered saline was added, and the tube was shaken vigorously by hand for 2 min. The sample was allowed to settle for 1 min, and 40 mL of the supernatant were filtered through a membrane filter. The filters were placed on membrane-*Enterococcus* indoxyl-β-d-glucoside medium (Becton, Dickinson and Company) and incubated at 41 °C ± 0.5 °C for 24 h. Then, 100 mL of seawater were passed through a membrane filter. Blue colonies growing on the filter after incubation were counted as *enterococci*. The coliform, *E. coli*, and *Enterococcus* counts in each sample were counted in triplicates, and the mean was calculated as the bacterial count (CFU/100 g or mL). The detection limits for coliforms, *E. coli*, and *Enterococcus* in the sand and water samples were 7 CFU/100 mL and 0.3 CFU/100 g, respectively.

### 2.3. Identification of Coliforms by Matrix-Assisted Laser Desorption Ionization Time-of-Flight Mass Spectrometry (MALDI-TOF MS)

MALDI-TOF MS (microflexLT/SH, Bruker Daltronics, Billerica, MA, USA) was used for genus identification [25]. Coliform strains isolated in BHI medium were spread thinly and evenly on a target plate. Following air-drying for 10 min, a template was overlaid with 1.0 μL of the matrix solution. All samples were analyzed using an Autoflex III TOF/TOF (Bruker Daltronics) operated in the linear positive mode within a mass range of 2000–20,000 Da based on the manufacturer’s instructions. For database construction and validation, measurements were performed in the auto-execute mode using Flex Control 3.4 software (Bruker Daltronics). The software settings were as follows: linear positive, 2–20 kDa; detector gain, 2691 V; laser shots, 40–200; and laser power, 30%. A Bruker bacterial test standard (part no. 8255343, Bruker Daltronics) was used for instrument calibration. Recorded mass spectra were analyzed using MALDI Biotyper Compass (Bruker Daltronics) under standard settings.

### 2.4. Continuous Flooding Simulation Using Beach Sand

The experiment was conducted under the assumption that dry sand would be moistened by rainfall and that the bacterial counts would change. Supratidal top surface sand at Kizaki Beach and Shirahama Beach was used for the simulated experiment. The characteristics of the sand were as follows: sunny for at least 3 weeks before collection, water content of 0.2–0.3%, surface temperature of 64 °C, and no coliforms or *E. coli* were detected. Each sediment sample (5 L) was placed in a plastic container (61.9 cm × 44.3 cm × 4.4 cm) disinfected with ethanol. Sterile distilled water was added evenly throughout the container until the sand was submerged to simulate rainfall. After water treatment, the containers were covered with a black plastic bag and shaded from light. Variations in coliform, *E. coli*, and *Enterococcus* counts were monitored every day for 2 weeks at room temperature (28 °C–31 °C). Approximately 5 g of sediment were randomly collected from three points and mixed in sterile plastic bags. Each fecal indicator bacterium was detected using the same method described for the sand collected from the beaches. Two sets of identical containers were prepared and tested in parallel. The average coliform, *E. coli*, and *Enterococcus* counts in each sand sample were calculated as the bacterial counts (CFU/100 g).

### 2.5. Estimation of Factors for Bacterial Growth

First, the effects of water content and temperature on the temporal change in the coliforms of all genera were analyzed using the state–space model as follows:(1)$$x_{t}=F_{t}x_{t-1}+\beta_{t}^{0}+\beta_{t}^{1}z_{t}^{1}+\beta_{t}^{2}z_{t}^{2}+G_{t}v_{t},\;v_{t}\sim N\left(0,Q_{t}\right)$$
(2)$$\log y_{t}=H_{t}x_{t}+w_{t},\;w_{t}\sim N\left(0,R_{t}\right)$$
where $x_{t}$ and $y_{t}$ represent the state values and observation values of the coliforms, respectively; $z_{t}^{1}$ and $z_{t}^{2}$ represent the water content and temperature at time t, respectively; $F_{t}$ is the link vector between the observed and latent variables; $H_{t}$ is the transition matrix describing the evolution of the state vector $x_{t}$; and $\beta_{t}^{0}$, $\beta_{t}^{1}$, and $\beta_{t}^{2}$ are parameters to be estimated.

Next, the effects of water content and temperature on the composition of bacteria were analyzed using the logistic regression model as follows:(3)$$y_{t}=\frac{1}{1+\exp\left\{-\left(\beta_{0}+\beta_{1}z_{t}^{1}+\beta_{2}z_{t}^{2}\right)\right\}}$$
where $y_{t}$ represents the ratio of the weight of a certain bacterium to the total weight of bacteria at time t; $z_{t}^{1}$ and $z_{t}^{2}$ represent the water content and temperature at time t, respectively; and $\beta_{0}$, $\beta_{1}$, and $\beta_{2}$ are parameters to be estimated.

## 3. Results and Discussion

### 3.1. Changes in the Fecal Indicator Bacterium Counts of Beaches

#### 3.1.1. Supratidal and Intertidal Top Surface Sand 

Figure 2a,b presents changes in daily precipitation, bacterial counts, and water content in supratidal and intertidal top surface sand at Kizaki Beach during the study period. In supratidal top surface sand, water content increased with rainfall, and the coliform counts increased rapidly by approximately 100 fold (26–2.23 × 10^3^ CFU/100 g) on the same day of rainfall or 1–2 days later (Figure 2a). For *E. coli* and *Enterococcus*, bacterial counts increased on the day of rainfall or 1 day later. The counts of *E. coli*, which were undetectable on sunny days, increased rapidly to 2.89 × 10^2^ CFU/100 g after rainfall. Conversely, when the weather recovered and the water content of the sand decreased, all bacterial counts declined. *E. coli* counts were considered more sensitive to environmental changes on the beach than coliform and *Enterococcus* counts because of the greater variability in detection and non-detection. A dependent relationship between *E. coli* counts and precipitation was observed, and *E. coli* counts in supratidal top surface sand were affected by a rainfall-dependent water supply. However, the counts of coliforms and *E. coli* in intertidal top surface sand ranged from 6.23 × 10^2^ to 3.7 × 10^3^ CFU/100 g and from 7 to 84 CFU/100 g, respectively, and they exhibited lower levels of variation than those observed in supratidal top surface sand (Figure 2b). Enterococcus counts increased rapidly on 14 June 2016, which was different from the changes in coliforms and *E. coli*. The water content of intertidal top surface sand was 9.90–30.3%, indicating that the sand was well supplied with water. Comparing supratidal and intertidal top surface sand, supratidal top surface sand displayed more variability in bacterial counts and showed a tendency to have higher peak bacterial counts. In particular, the counts of coliforms exceeded 1 × 10^3^ CFU/100 g in supratidal top surface sand.

Figure 3a,b presents changes in daily precipitation, bacterial counts, and water content in supratidal and intertidal top surface sand at Shirahama Beach during the study period. In supratidal top surface sand, bacterial counts increased with rainfall and decreased with decreasing water content, similar to the observations at Kizaki Beach. The peak counts for all bacteria were higher at Shirahama Beach than at Kizaki Beach; furthermore, these beaches showed dramatic increases and decreases in the peak counts. The water content of Shirahama Beach was higher than that of Kizaki Beach, and the water content exceeded 10% on multiple days. The sand at Kizaki Beach is homogeneous in terms of grain size (gravel: 0.0%, sand: 99.9%, silt: 0.1%, median particle diameter: 0.227 mm), and the main minerals included feldspar and quartz. Contrarily, Shirahama Beach contains large amounts of coral and shell fragments, and the grain sizes (gravel: 0.9%, sand: 98.7%, silt: 0.4%, median particle diameter: 0.280 mm) and shapes are heterogeneous. Differences in environmental conditions, including water content and porosity, of the two beaches may have influenced the variation in bacterial counts. *E. coli* was not detected in intertidal top surface sand on more days at Shirahama Beach than at Kizaki Beach. The counts of coliforms and *Enterococci* ranged from 1.6 × 10^2^ to 2.09 × 10^3^ CFU/100 g and from 23 to 6.57 × 10^3^ CFU/100 g, respectively, displaying greater variability than that observed at Kizaki Beach (Figure 3b). The water content of intertidal top surface sand at Shirahama Beach was always high, ranging from 11.3% to 72.2%, a condition under which the sand was submerged in water.

The investigations at Kizaki Beach and Shirahama Beach revealed that the bacterial counts in supratidal top surface sand were dependent on the water content of the sand attributable to precipitation, and they varied markedly depending on the drying of the sand and the water supply associated with rainfall. Additionally, these results implied the presence of a time lag between the supply of water and bacterial growth. The time from desiccation to the growth of bacteria in soil after water supply was reported to vary depending on the state of the bacteria [26]. In beach fecal bacteria, the variation in water content between dry and wet conditions might have a similar effect as that on soil bacteria. The temperature of the supratidal top surface sand fluctuated dramatically between 21.5 °C and 54.0 °C, even within a single day (Appendix A). Even at temperatures exceeding 50 °C, the bacteria in the sand remained viable and repopulated when the conditions were favorable.

#### 3.1.2. Supratidal and Intertidal 10 cm Sand 

We investigated the supratidal 10 cm sand at Kizaki Beach and Shirahama Beach, albeit with a lower frequency of investigations. Unlike top surface sand, *E. coli* was not detected in supratidal 10 cm sand at Kizaki Beach throughout the study period (Appendix A). *Enterococcus* was detected at counts of 42–67 CFU/100 g in two of the eight investigations. Coliforms were detected stably in the range of 31–2.56 × 10^2^ CFU/100 g. The bacterial counts and variability of each bacterium in supratidal 10 cm sand at Shirahama Beach were remarkably different from those in top surface sand (Appendix A). *E. coli* was detected at 9 CFU/100 g in two of the eight investigations. *Enterococcus* was prominently detected only on 6 June 2016 (7.86 × 10^2^ CFU/100 g). *Enterococcus* was detected at 9.91 × 10^2^ CFU/100 g in top surface sand on the same day, suggesting similarities in the variation of *Enterococcus* counts in the top surface and 10 cm sands. However, on days when *Enterococcus* was detected at 1.88 × 10^3^ CFU/100 g in top surface sand, its counts were as low as 8.0 CFU/100 g in 10-cm sand. Bacterial counts in the top surface sand and 10 cm sands, which had marked differences in terms of water content, temperature, and other variables, showed extremely prominent variations.

Appendix A show the changes in the bacterial counts and water content in intertidal 10 cm sand at Kizaki Beach and Shirahama Beach. The water content of this sand was >20%, and the sand was always wet. Furthermore, this sand from Kizaki Beach showed the presence of coliforms at constant counts ranging from 51 to 1.17 × 10^3^ CFU/100 g (Appendix A). *Enterococcus* counts varied widely in the range of 8.0–3.24 × 10^3^ CFU/100 g, with the day of the peak level coinciding with the peak count in intertidal top surface sand. Unlike the findings in supratidal 10 cm sand, *E. coli* was detected at counts of 9.0–4.36 × 10^2^ CFU/100 g in six of the eight investigations. The counts and changes in coliforms and *Enterococcus* were similar in intertidal 10 cm sand at Shirahama Beach (Appendix A). *E. coli* was detected at 10–4.43 × 10^2^ CFU/100 g. The day of the maximum peak count of *E. coli* at Kizaki Beach and Shirahama Beach did not coincide with the *E. coli* counts in top surface sand (top surface sand: 55, 33 CFU/100 g, 10 cm sand: 4.36 × 10^2^, 4.43 × 10^2^ CFU/100 g). The bacterial counts in intertidal 10 cm sand, which was immersed, fluctuated less frequently, thereby ensuring the steady presence of *E. coli* and *Enterococcus*. The bacterial counts and frequency of detection of fecal bacteria were higher in intertidal 10 cm sand than in supratidal 10 cm sand.

### 3.2. Fecal Indicator Bacteria in Seawater

Appendix A shows the coliform, *E. coli*, and *Enterococcus* counts in the seawater at Kizaki Beach during the investigation. The highest counts of coliforms, *E. coli*, and *Enterococcus* were detected on 6 June 2016 (6.1 × 10^3^, 1.03 × 10^2^, and 1.47 × 10^2^ CFU/100 mL, respectively). Before this date, rain fell for three consecutive days (4–6 June: 40.5, 24.5, and 73.5 mm, respectively). Conversely, coliforms, *E. coli*, and *Enterococcus* were detected at extremely low concentrations in seawater from 3 to 7 July 2016, when no precipitation was observed. These results indicated that fecal indicator bacteria in terrestrial or river water flow into coastal areas during rainfall, leading to seawater contamination. The main source of bacterial pollution at Kizaki Beach is believed to be the nearby Kiyotake River. Appendix A shows the coliform, *E. coli*, and *Enterococcus* counts in the seawater at Shirahama Beach during the investigation. The highest concentrations of coliforms, *E. coli*, and *Enterococcus* were detected on 6 June 2016, in line with the findings in seawater (3.5 × 10^3^, 25, and 1.72 × 10^2^ CFU/100 mL, respectively). Fecal indicator bacteria contained in land water or river water flowed into the coastal area during rainfall at Shirahama Beach. The main source of bacterial pollution at Shirahama Beach is thought to be the nearby Kaeda River. Fecal indicator bacteria in coastal seawater are thought to be supplied to supratidal top surface sand by high tide, waves, and wave breaking.

### 3.3. Changes in the Composition of Coliforms Isolated from Supratidal Top Surface Sand

In total, 549 coliforms isolated from supratidal top surface sand at Kizaki Beach during the investigation were identified via MALDI-TOF MS, and 22 bacterial genera were identified at this beach (Appendix A). Figure 4a shows the changes in coliforms in supratidal top surface sand during the investigation at Kizaki Beach. The coliforms in supratidal top surface sand had a completely different composition from the previous day, and daily fluctuations were identified. The dominant genera detected throughout the investigation were *Enterobacter* (252 strains), *Leclercia* (84 strains), *Citrobacter* (49 strains), *Lelliottia* (44 strains), and *Pantoea* spp. (42 strains). *Enterobacter* spp. are important causative agents of urinary tract infections, opportunistic infections, and pneumonia, and they should be considered from a public health perspective. Focusing on the rainy season from 25 June to 2 July 2016, when the coliform counts showed extensive variations, the proportions of *Leclercia* (2/3 strains) and *Enterobacter* spp. (3/5 strains) were high on 25 and 26 June, which were sunny days. From 27 to 30 June 2016, when rainfall continued, the proportion of *Enterobacter* spp. (53/100) was as high as on sunny days, and *Citrobacter* (20/100) and *Pantoea* spp. (11/100), which were not detected on sunny days, were also identified. However, on sunny days (1 and 2 July 2016), only *Enterobacter* spp. were identified (33/33 strains). 

In addition, changes in the composition of 745 coliforms isolated from supratidal top surface sand at Shirahama Beach were analyzed (Figure 4b). Similar to the findings at Kizaki Beach, the coliforms in supratidal top surface sand at Shirahama Beach varied dramatically on a day-to-day basis, and 25 bacterial genera were identified (Appendix A). Because different rivers flow into the coast, the bacterial species surviving in the sand are thought to show differences depending on the source of pollution. The most dominant species were *Enterobacter* spp. (328 isolates), accounting for 44% of all bacteria. *Pantoea* (79 isolates) and *Leclercia* spp. (73 isolates) were detected in large numbers. Focusing on the rainy season from 25 June to 2 July, *Acinetobacter* (23/42) and *Pantoea* spp. (13/42) were more abundant on 25 and 26 June when the weather was clear, differing form the findings at Kizaki Beach. Conversely, from 27 to 30 June 2016, when rainfall continued, the proportion of *Enterobacter* spp. was high (68/118 isolates), as observed at Kizaki Beach. A variety of bacteria, including *Cronobacter* spp. (13/118 strains), which were not detected on sunny days, were identified during rainfall. When rainfall subsided (1 and 2 July 2016), many *Klebsiella* spp. (23/30 strains) were observed on July 1, but none was detected the following day. Instead, *Pantoea* (16/25 strains) and *Enterobacter* spp. (7/25 strains) were dominant. We confirmed that the coliforms in the beaches were replaced by completely different bacteria in a single day. In addition, *Enterobacter* and *Citrobacter*, which can be pathogenic, propagated in the top surface sand of the beach in association with rainfall. 

Appendix A shows a list of diseases that can be caused by the bacterial genera identified on both beaches [27,28,29,30,31,32,33]. Beachgoers are at increased risk of gastrointestinal and diarrheal infections owing to contact with beach sand [22]. Therefore, beachgoers with weakened immune systems should be cautious when using beaches. A larger number of genera were identified in supratidal top surface sand at Shirahama Beach than at Kizaki Beach, suggesting that the source of contamination of coliforms differs between the beaches.

### 3.4. Bacterial Counts in Sand in a Continuous Flooding Simulation 

Figure 5a shows the *E. coli*, coliform, and *Enterococcus* counts in a continuous flooding simulation using sand from Kizaki Beach. After wetting the dry sand with distilled water, coliforms were detected at 47.5 CFU/100 g after 1 day, whereas they were undetectable before wetting. Furthermore, the coliform counts were 6-fold higher on day 5 than on day 1 (day 5: 3.08 × 10^2^ CFU/100 g). These results confirm that the coliform counts in the sand at Kizaki Beach are repopulated via the supply of water to the sand. By contrast, *E. coli* and *Enterococcus* were not detected throughout the investigation. This suggested that the repopulation of *E. coli* and *Enterococcus* is affected by factors other than water supply to the sand, such as the organic matter load. 

Figure 5b shows the *E. coli*, coliform, and *Enterococcus* counts in a continuous flooding simulation using sand from Shirahama Beach. The coliform counts ranged from 7.5 to 23.5 CFU/100 g, and coliforms either undetectable or close to the lower detection limit on days 1–4. However, the coliform count rapidly increased to 4.0 × 10^2^ CFU/100 g on day 5, as observed for Kizaki Beach. The in situ beach investigation revealed a rapid increase in the bacterial counts within 1–2 days, and the time to growth differed between the in situ investigation and the continuous flooding simulation. *E. coli* was detected only on day 5 (8.0 CFU/100 g). *Enterococcus* was detected at 14.5 CFU/100 g on day 3 and 80 CFU/100 g on day 5, but it was not detected on any other days. *E. coli* and *Enterococcus* were detected at low counts in the short term, and no clear growth trend was observed. These results indicate that coliform counts in sand are most easily increased by the supply of water to the sand, and coliforms have higher growth potential than *E. coli* and *Enterococcus*. The in situ beach investigation suggested that growth-limiting factors other than water supply attributable to rainfall exist for *E. coli* and *Enterococcus*. Each bacterium that was below the detection limit at the beginning of the experiment was increased by supplying water to the beach. This suggested that most of the bacteria on the beach were killed by the increase in beach surface temperature due to solar radiation, but some bacteria survived or were viable but non-culturable state.

### 3.5. Estimation of Factors for Bacterial Growth

Figure 6 and Figure 7 show the results of estimations of beach water content, sand surface temperature, and the constant term in the state–space model using data from 30 May to 6 July 2016, which were measured almost continuously at Kizaki Beach and Shirahama Beach. First, Kizaki Beach was considered. Figure 6a shows that the water content estimates were positive and significant, and coliform counts tended to increase as the water content increased. Water content in supratidal top surface sand was often high even 1 day after rainfall. As with seawater, supratidal top surface sand, which is directly encountered by beachgoers, exhibited increased bacterial counts after rainfall. Figure 6b shows that the temperature estimates were negative and significant, and coliform counts tended to decrease with increasing temperature. Unlike intertidal top surface sand, the surface temperature of supratidal top surface sand varied markedly with sunlight, and the beach surface temperature ranged from 22.0 °C to 54.0 °C during the investigation. In particular, the coliform counts were lower on days when the beach surface temperature exceeded 50 °C, well above the optimal growth temperature for bacteria in general, and a surface temperature of 20 °C–30 °C was optimal for coliform growth in supratidal top surface sand. Figure 6c shows that the constant term had little daily variation.

Considering Shirahama Beach, Figure 7a shows that the water content estimate was positive and significant, and, as noted for Kizaki Beach, the coliform counts tended to increase as the water content increased. Figure 7b shows that the estimated value of temperature was negative, meaning that the coliform count tends to decrease as the temperature increases. However, because the upper limit of the 95% confidence interval was positive on many days, it cannot be said that a statistically significant relationship existed. Figure 7c shows that the constant term had little daily variation. The sand surface temperature of Shirahama Beach was significantly lower than that of Kizaki Beach, but its water content was significantly higher (*p* < 0.01, Wilcoxon’s signed rank sum test). Shirahama Beach had a high water content ratio, which is a factor related to the coliform count, and the sand surface temperature was low. Thus, it can be said that the conditions were conducive to coliform growth. 

Appendix A shows the results of a logistic regression model in which the proportion of *Enterobacter* (Kizaki Beach: 45.9%; Shirahama Beach: 44.0%), which had the highest occupancy rate in both beaches, was used as the explained variable. The water content and temperature were not significantly different between the beaches, and it can be said that these variables had no effect on the proportion of *Enterobacter*. Therefore, it was inferred that factors other than the sand water content and temperature control the growth of *Enterobacter*. Just as diversity is lost when a few highly competitive species monopolize the resources necessary for growth and exclude less competitive species [34,35], the results suggest that *Enterobacter* is highly competitive among the bacteria at the beaches investigated in this study.

## 4. Conclusions

The counts of coliforms, *E. coli*, and *Enterococcus* in the top surface sand of coastal beaches were 4–100-fold higher within 2 days after rainfall than under sunny conditions (maximum counts: coliforms, 7.28 × 10^3^ CFU/100 g; *E. coli*, 1.54 × 10^3^ CFU/100 g; *Enterococcus*, 6.57 × 10^3^ CFU/100 g). The results of the investigations at Kizaki Beach and Shirahama Beach indicated that the respective bacterial counts in supratidal top surface sand were dependent on the water content of the sand attributable to precipitation, and their counts varied markedly with drying of the sand and the water supply associated with rainfall. The results also suggested a time lag between the supply of water and the growth of bacteria. Under conditions in which the water content of supratidal top surface sand was less than 5% and the temperature exceeded 50 °C, some of the bacteria in the sand remained viable and reproduced when conditions were favorable. The concentrations of each fecal indicator bacterium in coastal seawater also peaked after rainfall throughout the investigation, suggesting a supply of bacteria from seawater to sand. The coliform composition in supratidal top surface sand fluctuated dramatically, with *Enterobacter* spp. being the predominant species detected after rainfall. When the factors associated with bacterial growth were statistically estimated, it was confirmed that coliform growth in supratidal top surface sand depended on the water content and sand temperature. Because beachgoers inevitably make contact with the beach surface, it is suggested that rainfall supplies water to the dry sand, thereby increasing the counts of bacteria and the diversity of bacterial species, which might increase the health risk to humans. Information on bacterial contamination of the top surface sand of the supratidal zone after rainfall is of critical importance for improving public health. Information on factors promoting growth and the potential for fecal indicator bacterium repopulation should be accumulated. The monitoring of fecal indicator bacteria in both seawater and sand on coastal beaches should be continued with an emphasis on the relationship between bacterial growth and water supply attributable to rainfall. We propose the introduction of use regulations based on the bacterial contamination information obtained to improve public health at beaches, which are important recreational sites.

## Figures and Tables

**Figure 1 microorganisms-11-01074-f001:**
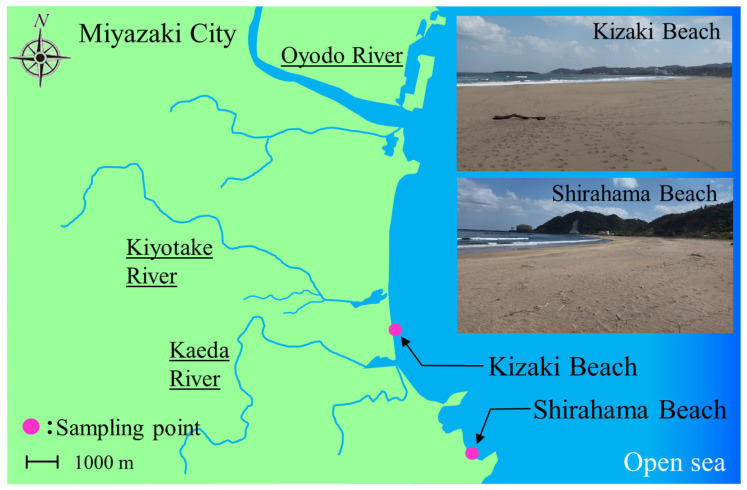
Coastal beaches, namely, Kizaki Beach (31°49′53.2″ N 131°27′08.8″ E) and Shirahama Beach (31°47′15.0″ N 131°28′53.5″ E), were sampled in this study.

**Figure 2 microorganisms-11-01074-f002:**
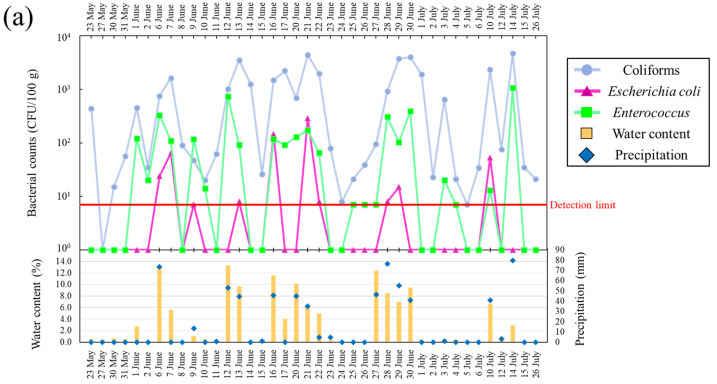
(**a**) Variation of *Escherichia coli*, coliform, and *Enterococcus* counts in supratidal top surface sand; water content; and precipitation at Kizaki Beach. (**b**) Variation of *Escherichia coli*, coliform, and *Enterococcus* counts in intertidal top surface sand; water content; and precipitation at Kizaki Beach.

**Figure 3 microorganisms-11-01074-f003:**
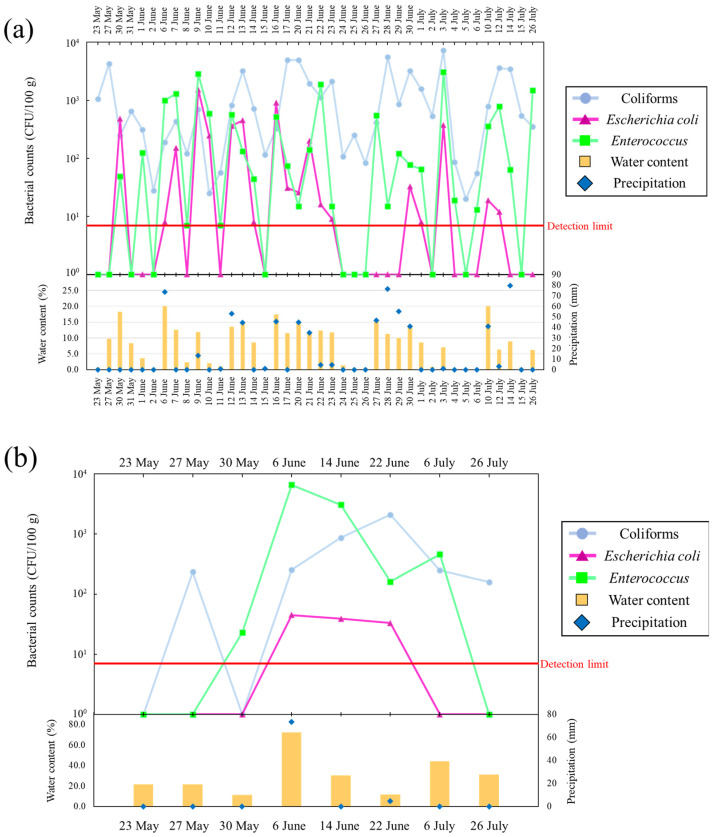
(**a**) Variation of *Escherichia coli*, coliform, and *Enterococcus* counts in supratidal top surface sand; water content; and precipitation at Shirahama Beach. (**b**) Variation of *Escherichia coli*, coliform, and *Enterococcus* counts in intertidal top surface sand; water content; and precipitation at Shirahama Beach.

**Figure 4 microorganisms-11-01074-f004:**
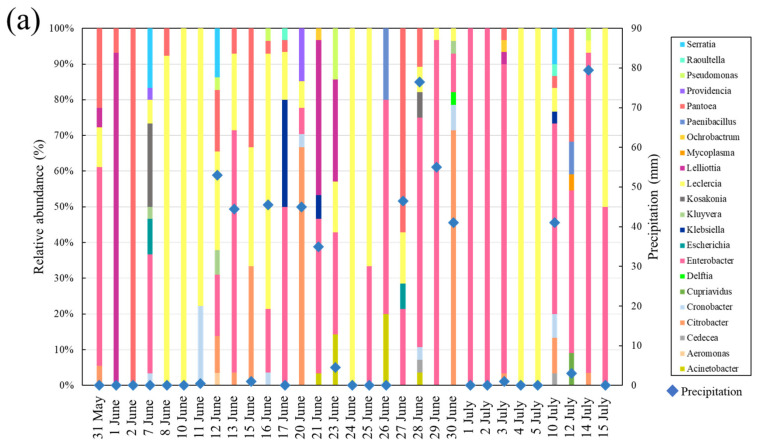
(**a**) Analysis of coliforms in supratidal top surface sand at Kizaki Beach. (**b**) Analysis of coliforms in supratidal top surface sand at Shirahama Beach.

**Figure 5 microorganisms-11-01074-f005:**
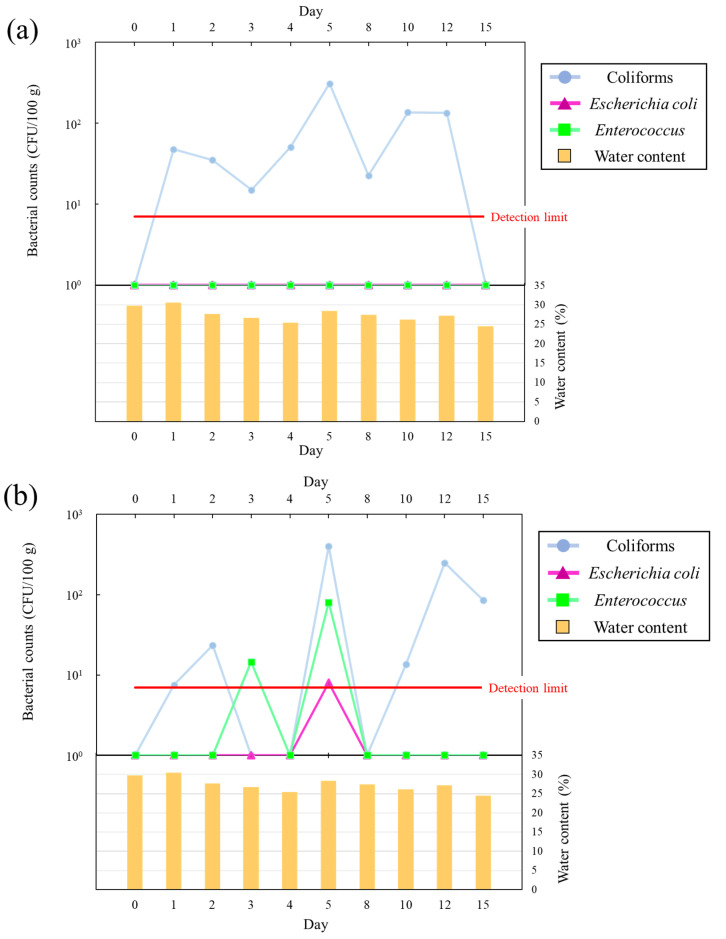
(**a**) Variation of *Escherichia coli*, coliform, and *Enterococcus* counts in a continuous flooding simulation of sand collected from Kizaki Beach. (**b**) Variation of *Escherichia coli*, coliform, and *Enterococcus* counts in a continuous flooding simulation of sand collected from Shirahama Beach.

**Figure 6 microorganisms-11-01074-f006:**
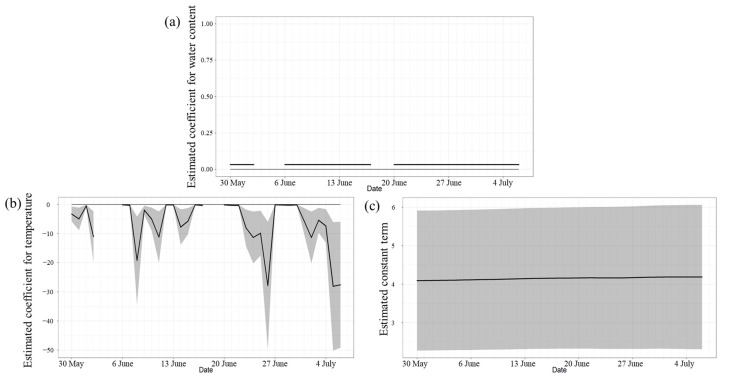
(**a**) Estimation of water content at Kizaki Beach. (**b**) Estimation of temperature at Kizaki Beach. (**c**) Estimation of the constant term at Kizaki Beach. The gray hatches in the figures indicate the 95% confidence intervals of the estimated values.

**Figure 7 microorganisms-11-01074-f007:**
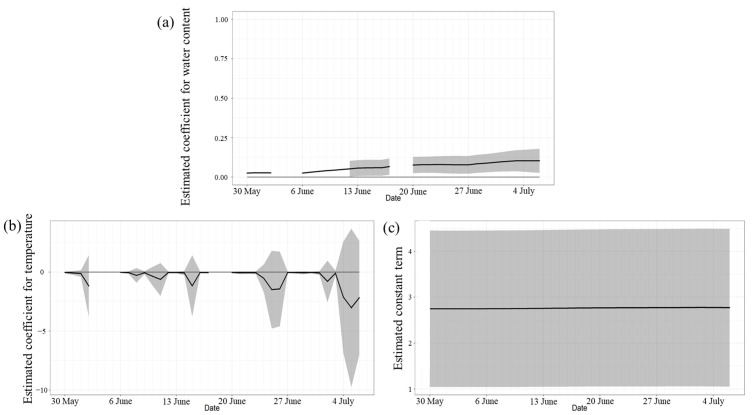
(**a**) Estimation of water content at Shirahama Beach. (**b**) Estimation of temperature at Shirahama Beach. (**c**) Estimation of the constant term at Shirahama Beach. The gray hatches in the figures indicate the 95% confidence intervals of the estimated values.

## Data Availability

Data will be made available on request.

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
