# Peer review of "Growth and Decay of Fecal Indicator Bacteria and Changes in the Coliform Composition on the Top Surface Sand of Coastal Beaches during the Rainy Season"

_microorganisms, 2023, doi:10.3390/microorganisms11041074_

Round 1

Reviewer 1 Report

This is an interesting study that will add to the body of literature.  The authors indicate clearly that indicators of fecal contamination have complicated cycles and we cannot simply rely on presence/absence of the indicators to draw conclusions to protect public health.

The manuscript is very descriptive, but I do not see metadata included.  I would love to know that the salinity levels were (and thus Aw which is key in the growth or regrowth of microorganisms).  Additionally temperature gradients may offer a better view of what is going on especially at different depths.  No data on UV radiation or pH are presented, and without these data it is rather hard to determine exactly what is going on.  The authors hint at the Viable-but-non-culturable state, but do not define it well.

I would be very happy to see the metadata included as part of the supplementary information.  These data will give a better understanding on what is going on.  I would imagine the air temperature as well as the radiation can be obtained from weather stations.  The salinity may also be key, since Enterococci seem to be better survivors under high salinity, not so with other indicators and that is why Enterococcus is the choice as an indicator for marine waters, whereas E. coli is used for fresh waters.

Please use italics when writing genus and species, this  throughout the manuscript.

All in all, the data presented are very interesting, but the devil is in the details, and there are many devils when carrying out environmental sampling.

Author Response

Thank you very much for providing important insights regarding our manuscript (ID, microorganisms-2329277). Your comments are very helpful in improving the accuracy and clarity of the manuscript. We have revised the manuscript in accordance with these suggestions. We are grateful for the time and energy you expended on our behalf.
Reply to comments and suggestions:
The revised points are shown in yellow highlight in the revised manuscript.
Comments and Suggestions for Authors
This is an interesting study that will add to the body of literature. The authors indicate clearly that indicators of fecal contamination have complicated cycles and we cannot simply rely on presence/absence of the indicators to draw conclusions to protect public health.
The manuscript is very descriptive, but I do not see metadata included. I would love to know that the salinity levels were (and thus Aw which is key in the growth or regrowth of microorganisms). Additionally temperature gradients may offer a better view of what is going on especially at different depths. No data on UV radiation or pH are presented, and without these data it is rather hard to determine exactly what is going on.
=> Salinity and pH
Measuring salinity and pH was difficult because we were targeting sands with low water content. As you indicated, salinity is an important factor controlling bacterial survival. In future studies, we will suspend sand samples in distilled water and measure the salinity and pH of the water.
=> Temperature gradient
In this study, since we focused on top surface sand (<1 cm), we did not measure subsurface temperature gradients, but evaluated only the temperature of the top surface sand. We have already investigated the bacteria in each layer of beach in previous study1.
In reference to your suggestion, we will also consider the subsurface temperature gradient in future investigations.
1. Suzuki, Y., Teranishi, K., Matsuwaki, T., Nukazawa, K. & Ogura, Y. Effects of bacterial pollution caused by a strong typhoon event and the restoration of a recreational beach: Transitions of fecal bacterial counts and bacterial flora in beach sand. Sci Total Environ 640-641, 52-61, doi:10.1016/j.scitotenv.2018.05.265 (2018).
=> We have added data for the sunshine duration, wind speed, and air temperature to Table S1 as you suggested. In this study, the tendency for bacterial counts to increase or decrease was confirmed by analyzing the temperature and water content of the sand as parameters.
[Line 76–77]
Table S1 shows data (sunshine duration, wind speed, precipitation, and air temperature) from meteorological stations near the sampling site.
The authors hint at the Viable-but-non-culturable state, but do not define it well.
=> In the experiment in Section 3.4, each bacterium was below the detection limit at the beginning of the experiment, but each bacterium increased with the supply of water to the sand. This suggested that most of the bacteria on the beach were killed by the increase in beach temperature due to solar radiation, but some survived or were in a Viable-but-non-culturable state.
We have described this fact more clearly.
[Line 376–380]
Each bacterium that was below the detection limit at the beginning of the experiment was increased by supplying water to the beach. This suggested that most of the bacte-ria on the beach were killed by the increase in beach surface temperature due to solar radiation, but some bacteria survived or were Viable-but-non-culturable state.

I would be very happy to see the metadata included as part of the supplementary information. These data will give a better understanding on what is going on. I would imagine the air temperature as well as the radiation can be obtained from weather stations. The salinity may also be key, since Enterococci seem to be better survivors under high salinity, not so with other indicators and that is why Enterococcus is the choice as an indicator for marine waters, whereas E. coli is used for fresh waters.
=> We have added data for the sunshine duration, wind speed, and air temperature to Table S1.
Unfortunately, as mentioned above, we did not measure salinity.
Please use italics when writing genus and species, this throughout the manuscript.
=>We have corrected the genus and species to italicize.
All in all, the data presented are very interesting, but the devil is in the details, and there are many devils when carrying out environmental sampling.
=>We will keep in mind the points you have indicated and proceed with future study.

Reviewer 2 Report

The article is interesting and provides valuable information on the factors controlling the growth of bacteria in the sand of seaside beaches. It has a scientific and cognitive character. The article has a correct structure, is clearly written, introduction and review of literature sources at a sufficient level.

I have one remark and I ask that the authors consider placing a table with a list of identified bacteria and diseases that can be caused by these bacteria. Not everyone knows how dangerous microorganisms living in sand on beaches can be. This information is important for the protection of public health.

Author Response

Thank you very much for your suggestions regarding our manuscript (ID, microorganisms-2329277).
Your comments are very helpful for our revised manuscript. We have revised the manuscript in
accordance with these suggestions.
Reply to comments and suggestions:
The revised points are shown in yellow highlight in the revised manuscript.
Comments and Suggestions for Authors
The article is interesting and provides valuable information on the factors controlling the growth of
bacteria in the sand of seaside beaches. It has a scientific and cognitive character. The article has a
correct structure, is clearly written, introduction and review of literature sources at a sufficient level.
=> We thank the reviewer for these excellent comments.
I have one remark and I ask that the authors consider placing a table with a list of identified bacteria
and diseases that can be caused by these bacteria. Not everyone knows how dangerous
microorganisms living in sand on beaches can be. This information is important for the protection
of public health.
=> In accordance with your suggestion, we have summarized the cases of infections of the major
bacterial genera in Table S6.
[Line 341–342]
Table S6 shows a list of diseases that can be caused by the bacterial genera identified on both
beaches.

Round 2

Reviewer 1 Report

A few grammatical errors which, I am sure, will be caught by the editors.  This is an improved manuscript and should be ready to be accepted.

Author Response

We appreciate your important remarks prior to the publication of our paper. The revised paper was checked and edited by an English proofreading company.  Our paper is now finished.
